# Wafer-sized multifunctional polyimine-based two-dimensional conjugated polymers with high mechanical stiffness

Hafeesudeen Sahabudeen[1], Haoyuan Qi[2], Bernhard Alexander Glatz[3], Diana Tranca[4], Renhao Dong[1], Yang Hou[1], Tao Zhang[1], Christian Kuttner[3], Tibor Lehnert[2], Gotthard Seifert[4], Ute Kaiser[2], Andreas Fery[1,3], Zhikun Zheng[1] & Xinliang Feng[1]

One of the key challenges in two-dimensional (2D) materials is to go beyond graphene, a prototype 2D polymer (2DP), and to synthesize its organic analogues with structural control at the atomic- or molecular-level. Here we show the successful preparation of porphyrin-containing monolayer and multilayer 2DPs through Schiff-base polycondensation reaction at an air–water and liquid–liquid interface, respectively. Both the monolayer and multilayer 2DPs have crystalline structures as indicated by selected area electron diffraction. The monolayer 2DP has a thickness of $\sim 0.7$ nm with a lateral size of 4-inch wafer, and it has a Young's modulus of $267 \pm 30$ GPa. Notably, the monolayer 2DP functions as an active semiconducting layer in a thin film transistor, while the multilayer 2DP from cobalt-porphyrin monomer efficiently catalyses hydrogen generation from water. This work presents an advance in the synthesis of novel 2D materials for electronics and energy-related applications.

[1] Center for Advancing Electronics Dresden (cfaed) and Department of Chemistry and Food Chemistry, Dresden University of Technology, Mommsenstraße 4, 01062 Dresden, Germany. [2] Central Facility for Electron Microscopy, Department of Electron microscopy of Material Science, University of Ulm, 89081 Ulm, Germany. [3] Leibniz-Institut für Polymerforschung Dresden e.V., Institute of Physical Chemistry and Polymer Physics, Department of polymer interface, Hohe Straße 6, 01069 Dresden, Germany. [4] Department of Theoretische Chemie, Dresden University of Technology, Bergstraße 66b, 01062 Dresden, Germany. Correspondence and requests for materials should be addressed to Z.Z. (email: zhikun.zheng@tu-dresden.de) or to X.F. (email: xinliang.feng@tu-dresden.de).

Two-dimensional (2D) polymers (2DPs) that are laterally infinite, one atom- or monomer-unit thin, free-standing, covalent networks with long-range order along two orthogonal directions have attracted intense attention in recent years due to their wide applications in electronics, membrane and sensing[1–4]. A prototypical example of the 2DPs is graphene, a zero bandgap semiconductor consisting of an atomic layer of sp[2]-carbons, which has demonstrated exceptional electron mobility, quantum Hall effect and ballistic charge carrier transport behaviour[5]. However, if not provided by nature, the synthesis of graphene involves high-energy procedures, such as chemical vapour deposition (CVD), epitaxial growth and pyrolysis[6,7]. The harsh experimental conditions preclude molecular design of graphene on demand. As a first step to rationally synthesize 2DPs, 2D monolayers of porphyrin nanostructures and porous graphene on crystalline metal surfaces have been constructed through Ullmann coupling under ultra-high vacuum conditions[8,9]. However, the synthesized 2D monolayers are limited to nanometers in size, and their release from original substrate is challenging. As an alternative approach, 2DPs have been recently achieved through ultraviolet initiated cycloaddition of anthracene-based monomers either in lamellar organic crystals followed by exfoliation into individual layers or at an air–water interface of a Langmuir-Blodgett (LB) trough[10–13]. The exfoliation method provides synthetic 2DPs with limited lateral size, typically ranging from a few hundred nanometers to several micrometers, which in the end are determined by crystal sizes. In contrast, the air–water interface method in principle offers 2DPs with unlimited lateral size, which is confined to that of irradiation spot for photochemical reaction. Despite the current progress, conjugation is missing in synthetic 2DPs, which impedes their applications in electronics and optoelectronics. Moreover, the mechanical property and functionality of synthetic 2DPs have not been explored so far. The future research and application of this intriguing class of materials urgently call for the efficient synthesis of 2DPs with versatile methodology and structures.

Schiff-base condensation reaction, which forms a conjugated imine bond from an amine and an aldehyde group has been widely explored in the synthesis of covalent organic frameworks (COFs)[14,15]. Especially, intramolecular hydrogen bonds formed at the imine center can significantly enhance the structural rigidity, crystallinity and chemical stability of the synthesized COFs in aqueous solution[16]. Thus, we envision that such a reaction will be a promising methodology for the synthesis of large-area 2DPs at the water-containing interfaces. Although Schiff-base reaction without intramolecular hydrogen bond formation has been recently applied to obtain organic thin layers at an air–water interface, they have no proven crystallinity, which is a key feature regarding the synthesis and application of 2D layered materials[17]. Herein, we demonstrate the efficient synthesis of crystalline monolayer and multilayer 2DPs through Schiff-base condensation reaction between 5,10,15,20-tetrakis(4-aminophenyl)-21H,23H-porphine (monomer 1) or 5,10,15,20-tetrakis(4-aminophenyl)-21H,23H- porphyrin-Co(II) (monomer 2) and 2,5-dihydroxyterephthalaldehyde (monomer 3) at an air–water and water–chloroform interface, respectively. The resulting monolayer 2DP (4), which is as large as a 4-inch wafer, has demonstrated outstanding mechanical robustness with a Young's modulus of $\sim 267 \pm 30$ GPa. Remarkably enough, the achieved monolayer 2DP (4) can function as an active semiconducting layer in a thin film transistor (TFT), whereas the multilayer 2DP (5) from cobalt-porphyrin monomer can efficiently catalyse the hydrogen generation from water.

## Results

**Synthesis and morphology of monolayer 2DP.** Figure 1 illustrates the interfacial synthesis of targeted 2DPs by Schiff-base polycondensation. A sub-monolayer of monomer 1 was first spread at an air–water interface of a LB trough from chloroform solution and compressed to a surface pressure of 10 mN m$^{-1}$, which promoted the formation of densely packed 1. Afterwards, monomer 3 was added to the subphase and diffused to the interface where 2D polymerization was triggered by the formation of imine bonds between amine and aldehyde groups in monomer 1 and 3, respectively. The interfacial reaction has been kept for more than four hours (vide infra) to promote the reaction conversion for the formation of the 2DP (4).

The resulting 2DP (4) can be readily transferred to any substrates. For instance, the horizontal transfer of the 2DP from the top of the water surface onto holey copper grid led to a freely suspended film over hexagonal holes with a slide length of $\sim 20 \mu m$ (Fig. 2a). A few cracks and folds were observed which are likely due to the mechanical stain caused during the transfer or/and drying process. Nevertheless, continuous film with an area of 1 mm$^2$ was typically obtained, suggesting high mechanical stiffness of the 2DP. Control experiments under the same experimental conditions for the synthesis of 2DP involving only monomer 1 or 3 failed to obtain a freestanding film.

After a vertical transfer of the 2DP onto 300 nm SiO$_2$/Si wafers, optical microscopy (OM) and atomic force microscopy (AFM) were performed to study the morphology of the 2DP. The OM image shows a macroscopically homogenous film with long straight edges (several hundred micrometers, Fig. 2b), indicating that the 2DP breaks along a cleavage direction. The crack helps to locate an AFM tip to measure the film thickness, which amounts to $\sim 0.7$ nm (Fig. 2c), thus suggesting the single-layer feature of the synthesized 2DP.

**Structural characterization of mono- and multi-layer 2DPs.** The internal structure of the monolayer 2DP was investigated by selected area electron diffraction (SAED) performed in a transmission electron microscope (TEM). However, due to the high electron beam sensitivity the freestanding 2DP was deteriorated rapidly and no diffraction pattern could be observed even under cryogenic condition (− 175 °C). To combat the negative effects of electron radiation, such as knock-on damage, electrostatic charging and chemical etching, a graphene encapsulation method was applied to enhance the radiation resistance of the 2DP (ref. 18). To this end, a graphene-2DP-graphene sandwich structure was constructed using graphene grown on copper foil by CVD at $\sim 1,000$ °C. In addition, low dose (below 0.05 e$^-$ Å$^{-2}$ s$^{-1}$) diffraction technique was applied to better preserve the structure of the 2DP. Several rings composed of weak but clear diffraction spots were observed from the sandwich structures (Fig. 2d, Supplementary Fig. 1). The diffraction spots at 4.7 and 8.1 nm$^{-1}$ correspond well to the first and second order reflections of graphene[19]. The other diffraction spots originate from the 2DP, proving the existence of long-range order within the synthesized monolayer film. The diffraction spots at 2.7 nm$^{-1}$ ($\sim 0.37$ nm) and 4.5 nm$^{-1}$ ($\sim 0.22$ nm) are attributed to the cavity of the porphyrin units ($\sim 0.4$ nm) and benzene and dihydroxylbenzene rings, respectively. This result suggests that the monolayer film consists of highly ordered arrangement of monomers 1 and 3, which agrees well with the structure of the 2DP based on density functional based tight binding (DFTB) calculation (Fig. 2e). It is worth mentioning that the same diffraction rings has been repeatedly observed from different positions of the monolayer, which indicates high homogeneity of the polycrystalline 2DP on the unit-cell level (Supplementary Fig. 1).

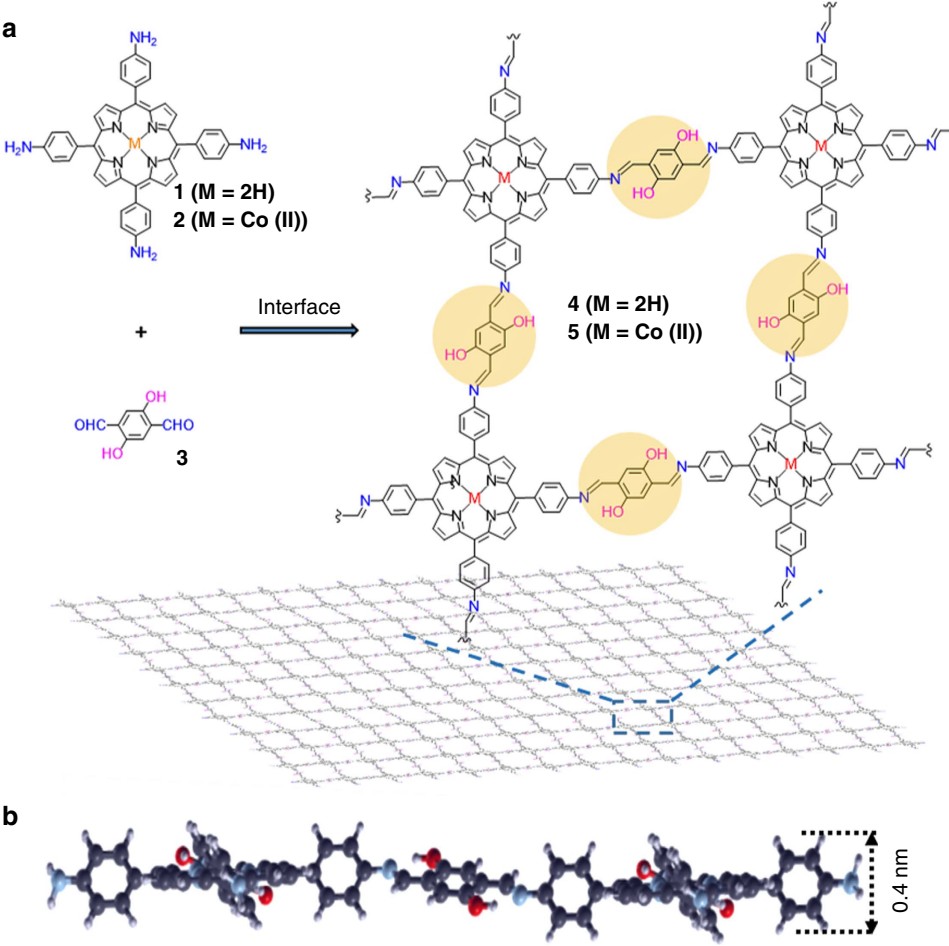

**Figure 1 | Synthesis of a 2DP through Schiff-base condensation reaction at an interface.** (**a**) Chemical structures of monomers (**1**, **2** and **3**) and 2DPs (**4** and **5**). (**b**) Cross-sectional view of the molecular structure of a monolayer 2DP (**4**) suggested by DFTB.

AFM, TEM and SAED measurements have also been performed on a multilayer 2DP film, which was prepared at liquid–liquid interface using an aqueous solution of monomer **3** (upper phase, $1 \times 10^{-3}$ mol l$^{-1}$) and a chloroform solution of monomer **1** (lower phase, $2 \times 10^{-4}$ mol l$^{-1}$) with a reaction time of 10 days (Supplementary Fig. 2). The synthesized multilayer was fished onto 300 nm SiO$_2$/Si, which gave an AFM thickness of ∼20 nm (Fig. 2f). It could also be fished onto a copper grid, leading to a freely suspended film, and was further characterized with TEM and SAED. At the edge of the film, laminar structures were identified, suggesting that the synthesized 2DP had a multilayer feature (Fig. 2g). High-resolution image of the film exhibits well-ordered lines due to the stacking of the layers. The layer distance is about 0.4 nm, pointing out the thickness of each layer, which is similar to that predicted by DFTB calculation (Fig. 1b). When SAED was performed on the film, moiré fringes were observed, which indicated that the multilayer 2DP adopted slight interlayer offsets typical for stacked aromatic systems that have been observed within other layered materials, such as trilayer graphene and exfoliated 2D COFs[20,21]. The shape of the moiré fringes is square (Fig. 2h, inset), indicating the lattice geometry for the 2DP, which is in agreement with the structure derived from DFTB calculation (Fig. 2e).

**Spectroscopic characterization of 2DPs.** To evaluate the chemical composition and homogeneity of the 2DP, the monolayer was horizontally transferred onto a quartz substrate and characterized by ultraviolet–visible spectroscopy. Ultraviolet–visible spectra of the monolayer shows the characteristic Soret (S) band at 442 nm and Q bands at 526, 571 and 656 nm, which correspond to porphyrin units, thereby further confirming their presence in the 2DP (Fig. 3a). The S band of the 2DP red-shifts by 20 nm in comparison to that of the monomer **1** (Supplementary Fig. 3a), suggesting the existence of extended conjugation within the 2DP. Assuming that all the porphyrin moieties were lying parallel to the surface, the measured absorbance (∼0.05) of the S band from the 2DP agrees well with calculated value of a monolayer with 100% surface coverage. A multilayer 2DP was fabricated by repeated horizontal transfer of monolayers on a quartz substrate in a layer-by-layer fashion to evaluate the homogeneity of the 2DP and study the change in optical properties with controlled growth of layer thickness. Figure 3b shows a linear correlation between the absorbance value of the S band and the number of layers, following the Beer–Lambert law. The result clearly shows that the addition of each layer leads to the deposition of same amount of porphyrin units, which confirms the macroscopic homogeneity of the 2DP. The higher signal-to-noise ratio of the multilayer in comparison with that of monolayer also gives a more precise estimation of the optical band gap of the 2DP from the Q band of the absorbance spectra, which amounts to 1.4 eV (Supplementary Fig. 3b). The electronic band gap of the 2DP is also computed by DFTB, and the corresponding projected density of states indicate a value of 1.3 eV (Fig. 3c). Both the experimental and theoretical values are

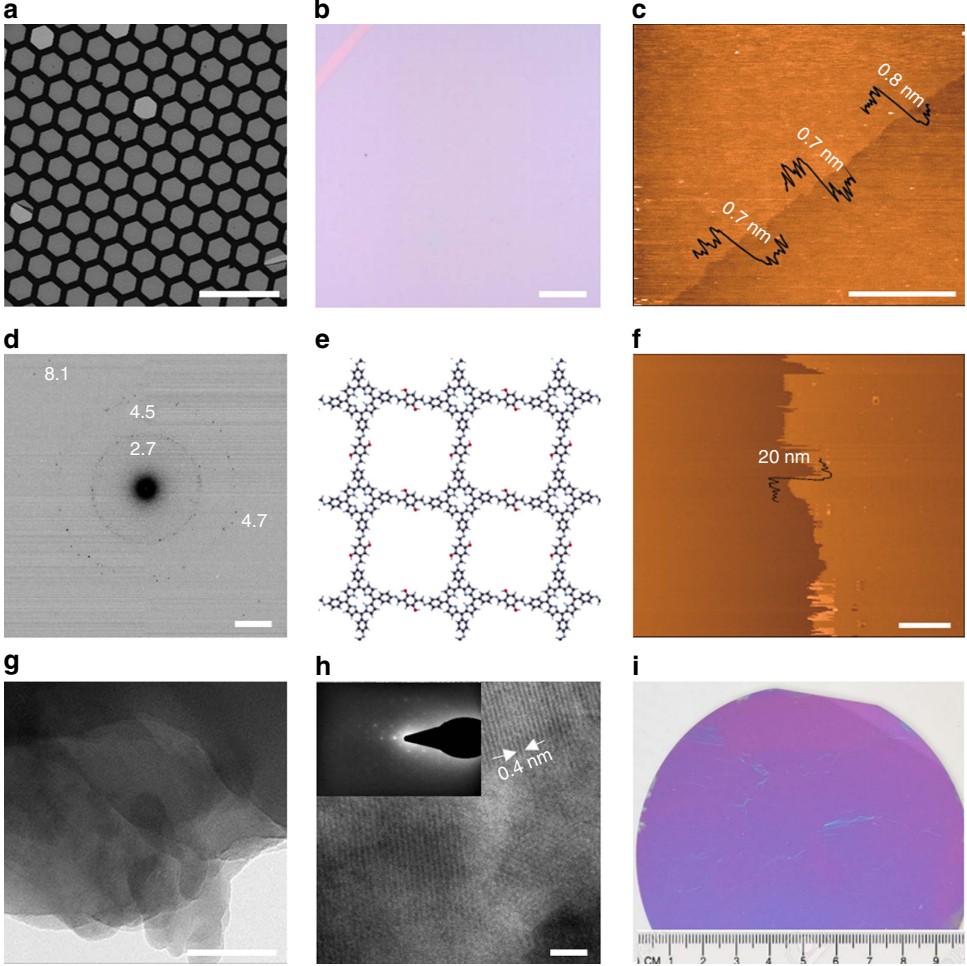

**Figure 2 | Morphology and structural characterizations of 2DP (4).** (**a**) Scanning electron and (**b**) optical microscopy images of monolayer 2DP suspended over a copper grid and deposited on 300 nm $SiO_2$/Si, respectively. (**c**) Atomic force microscopy (AFM) image of the monolayer 2DP on 300 nm $SiO_2$/Si. (**d**) Selected area electron diffraction (SAED) pattern of the monolayer 2DP sandwiched by two layers of graphene (G), G/2DP/G. (**e**) Molecular structure of the monolayer 2DP predicted by DFTB calculation. (**f**) AFM, and (**g**,**h**) TEM images of a multilayer 2DP synthesized at liquid–liquid interface. The insert in **h** is a SAED pattern of the multilayer. (**i**) Photographic image of monolayer 2DP on 4-inch 300 nm $SiO_2$/Si wafer. Scale bar, 100 μm (**a**); scale bar, 100 μm (**b**); scale bar, 3 μm (**c**); scale bar, 2 nm$^{-1}$ (**d**); scale bar, 3 μm (**f**); scale bar, 50 nm (**g**) and scale bar, 3 nm (**h**), respectively.

consistent with each other, which supports the semiconducting nature of the synthesized 2DP.

To gain insights into the chemical composition of the monolayer as well as the multilayer 2DPs, they were horizontally transferred onto 300 nm $SiO_2$/Si and Au/Si, and characterized by confocal Raman spectroscopy and X-ray photoelectron spectroscopy (XPS). Raman spectroscopy has been shown to have a sensitivity down to the single-molecule level and is capable of monitoring ions evolution in monolayer sheet[22–24], which is vital for the characterization of 2DPs. Thus, we investigated the Raman spectra of monomers (**1**, **3**) as well as the monolayer and multilayer 2DPs (**4**) by measuring variations in bands corresponding to the amine, aldehyde and imine groups (Fig. 3d). The Raman spectra show the N-H bending band at 1,280 cm$^{-1}$ for monomer **1** and the aldehyde C=O stretch at 1,675 cm$^{-1}$ for monomer **3**, respectively[25,26]. Both bands disappear in the spectrum of the monolayer 2DP (**4**), revealing that the amine and aldehyde groups have been successfully transformed in **4**. The spectra of the monolayer and multilayer 2DPs have similar features, indicating that there are no significant difference in their chemical bonding and compositions (Supplementary Fig. 4). In contrast to the monomers, a new band at 1,593 cm$^{-1}$ characteristic for –C=N stretching is observed for 2DPs,

highlighting the formation of imine bonds. The presence of this characteristic band provides a chance to use its intensity to qualitatively monitor the reaction time for the formation of the monolayer 2DP. In this aspect, time-dependent experiments were performed for the polycondensation reaction at the air–water interface with a period of 0.5, 1, 2, 4, 8 and 16 h. No significant change in the intensity of imine band was identified after 4 h, suggesting the efficient conversion for the formation of the 2DP (Supplementary Fig. 5).

The XPS spectra of the monolayer and multilayer 2DPs are similar to each other, further confirming no obvious difference in their chemical compositions (Supplementary Fig. 6). The N1s signal of the 2DP has two peaks with binding energies at 400 and 398.4 eV, which are attributed to pyrrole nitrogen (-NH-) and imine nitrogen with hydrogen bond (O-H···N=C) (blue), and imine nitrogen in porphyrin units (red), respectively (Fig. 3e)[27]. The ratio of intensities of the blue to red peaks at higher to that of lower binding energies is 3:1, which manifests that each porphyrin monomer **1** forms four imine bonds with monomer **3** in 2DP. The O1s signal is much stronger than expected from the chemical structure of **4**, which is caused by the existence of large amount of water associated with interfacial synthesis procedure (Supplementary Fig. 6). A quantitative analysis of the

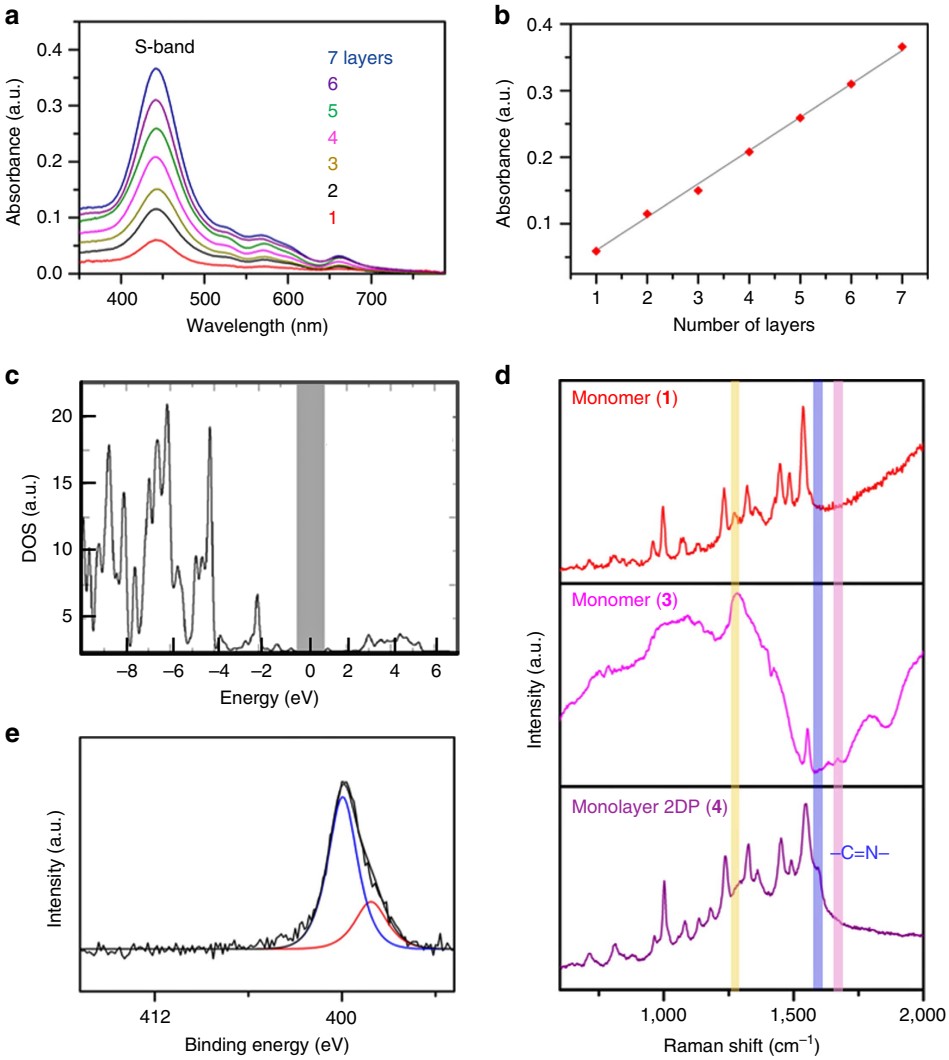

**Figure 3 | Spectroscopic characterizations of 2DP (4).** (**a**) Ultraviolet–Vis spectra of up to seven layers of 2DPs on quartz. (**b**) Plot of S band absorbance versus number of 2DPs. (**c**) Density of states (DOS) versus energy of the 2DP as calculated by DFTB. (**d**) Raman spectra of monomers (**1**, **3**) and 2DP (**4**) on 300 nm $SiO_2$/Si. (**e**) X-ray photoelectron spectrum for N1s signal of monolayer 2DP on Au/Si. a.u., arbitrary unit.

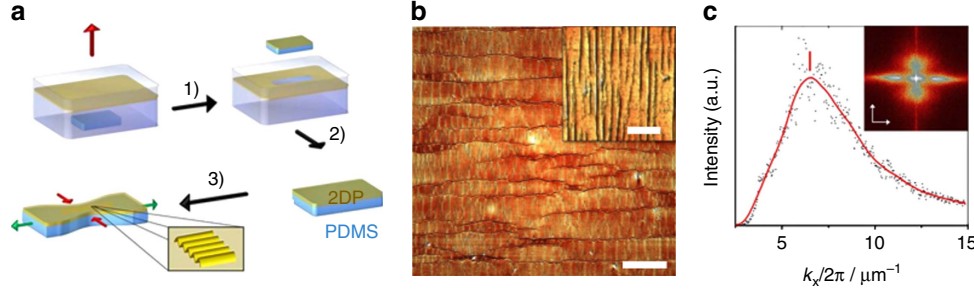

**Figure 4 | Mechanical characterization of monolayer 2DP (4).** (**a**) Schematic depiction for the synthesis of the 2DP (1), transfer (2) onto a plain elastomeric substrate (PDMS), and transversal compression upon uniaxial lateral stretching of the 2DP on PDMS (3), inducing wrinkling for the monolayer. (**b**) AFM topography image of the wrinkled 2DP. The insert in **b** shows an amplified image of the wrinkles. (**c**) Integrated intensity profile after 2D Fourier-transformation (see insert) along $k_x$ direction revealing a mean periodicity of 152 nm. Scale bars in **b** and insert are 5 µm and 500 nm, respectively. a.u., arbitrary unit.

N1s and C1s signals suggests an $N/C_{aromatic}$ ratio ($\sim$ 15:100) similar to the expected chemical composition of **4** (N/C = 14:100) (Supplementary Fig. 6). In conjunction with the Raman and XPS spectra, our results strongly validate the formation of imine 2DP with high efficiency at an air–water interface.

**Lateral size of monolayer 2DP.** The high reaction conversion for the formation of 2DP (**4**) triggers our curiosity to know how large its lateral size can be, which has been demonstrated by a horizontal transfer onto a 4-inch-sized 300 nm $SiO_2$/Si wafer partially covered by a piece of mica. Mica removal by tweezers

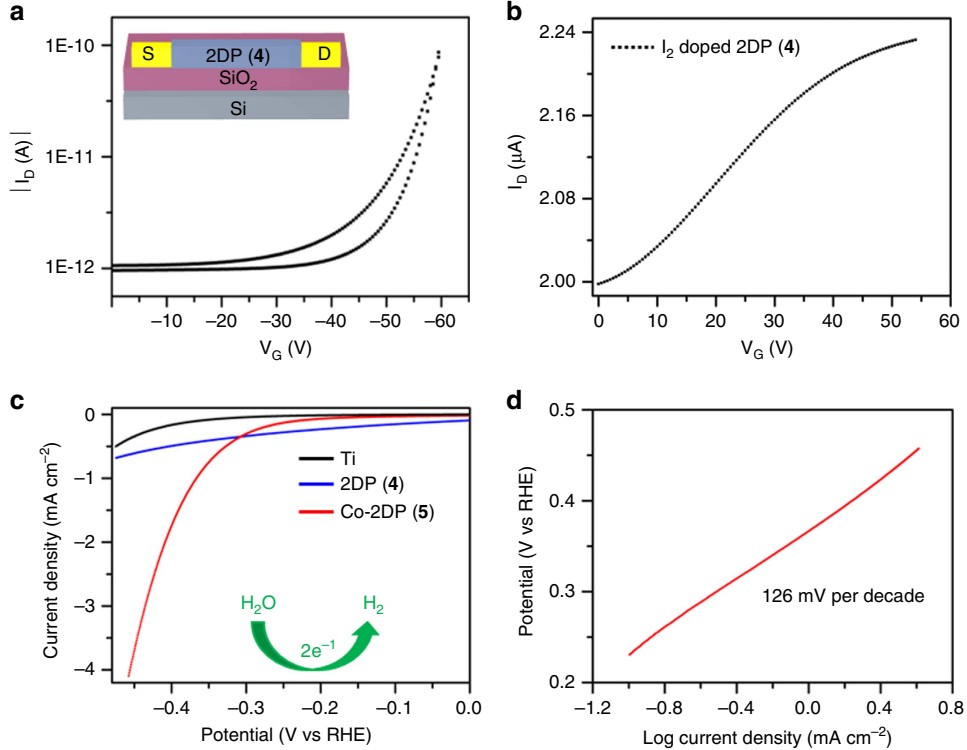

**Figure 5 | Application of 2DPs in thin film transistors and for H$_2$ generation from water.** Transfer curve of a thin film transistor employing 2DP (**4**) as an active semiconducting layer at a source to drain voltage of − 40 V (**a**) before and (**b**) after doping with iodine. (**c**) Hydrogen evolution reaction polarization plots of blank titanium foil (black), multilayer 2DP (**4**) (blue) and cobalt-2DP (**5**) (red) at a scan rate of 5 mV s$^{-1}$ in 1.0 mol l$^{-1}$ KOH. (**d**) Tafel plot of cobalt-2DP (**5**) with a slope of 126 mV decade$^{-1}$.

breaks the monolayer and leads to a several cm$^2$-sized crack within the 2DP, which gives obvious contrast and enables clear observation of the wafer-sized 2DP by the eyes or photography (Fig. 2i, Supplementary Fig. 7).

**Mechanical stiffness of monolayer 2DP.** To determine the mechanical properties of the monolayer 2DP, a Strain-Induced Elastic Buckling Instability for Mechanical Measurements technique has been employed[28]. For the measurement, the monolayer was horizontally transferred onto an elastomeric support—polydimethylsiloxane (PDMS). Special care was taken to ensure a strong adhesion of the film on the elastomer, which was achieved by a drying process. The resulting sample was compressed, which led to the formation of a regular wrinkling pattern in the 2DP as recorded by AFM (Fig. 4a,b). Perpendicular to the wrinkling pattern, elongated cracks were observed, which traced back to the pre-strain applied to the 2DP during the compression of the sample. These cracks did not affect the wrinkling periodicity and thus could be omitted in the mechanical analysis. Correlation between the wavelength of the wrinkling and thickness of the monolayer and the mechanical stiffness of the PDMS allows the evaluation of the Young's modulus for the 2DP (Supplementary Methods)[29,30]. A 2D Fourier-transformation and integration along wrinkling direction ($k_x$) were performed to estimate the wavelength of the wrinkling, which gave the first order peak at 6.57 µm$^{-1}$ (Fig. 4c). This value corresponds to a wavelength of 152 nm, which is similar to that obtained using a complementary analysis technique based on the power spectral density (Supplementary Fig. 8). Accepting the AFM height of 0.7 nm as the monolayer thickness, an average Young's modulus of $E_{Young} = 267 \pm 30$ GPa is obtained (Supplementary Figs 8 and 9). This value is comparable to the $E_{Young}$ of other crystalline 2D sheets, such as graphene (1 TPa)

(ref. 31) and graphene oxide (208 GPa) (ref. 32), and is much higher than $E_{Young}$ of organic monolayers with ∼ 66% reaction conversion, such as bis(terpyridine)-metal complex nanosheet (16 GPa) (ref. 33) and cycloaddition-induced anthracene nanosheet (11 GPa) (ref. 34), which reflect the high crystallinity and mechanical stiffness of the achieved monolayer 2DP in this work.

**Prototype applications of monolayer and multilayer 2DPs.** Next, the monolayer 2DP has been integrated in a thin-film transistor for a proof-of-concept demonstration of the synthesized 2D materials as an active semiconducting layer. The 2DP was horizontally transferred onto an n-doped Si wafer with 300 nm SiO$_2$ (dielectric), on which 30 nm thick gold was deposited and used as bottom electrodes. From the transfer curve of the 2DP (Fig. 5a), a mobility of $1.3 \times 10^{-6}$ cm$^2$ V$^{-1}$ S$^{-1}$ and an on/off ratio of 10$^2$ were obtained. When the 2DP was doped by I$_2$, the charge carrier mobility increased by a factor of more than two orders of magnitude and reached $1.6 \times 10^{-4}$ cm$^2$ V$^{-1}$ S$^{-1}$ (Fig. 5b). These results clearly manifest that the synthetic 2DP has the potential to be used as semiconducting material for TFTs under ambient conditions.

Cobalt porphyrin in which a cobalt atom is coordinated with four nitrogen atoms (Co-N$_4$) can serve as active sites of molecular catalysts for electrocatalytic hydrogen evolution reaction (HER)[35] in aqueous solution. However, the molecular catalysts need to be immobilized onto electrode in a well-defined manner in order to avoid low catalytic activity and structural instability[36,37]. Given that cobalt-porphyrin can be well-positioned in a large-area 2DP with high-density, the resulting film can be directly deposited onto an electrode surface thus functioning as a catalyst for HER. Hence, a multilayer 2DP (**5**) by reacting 5,10,15,20-tetrakis (4-aminophenyl)-21*H*,23*H*- porphyrin-Co(II) (**2**) with monomer

(3) was constructed at water–chloroform interface. Subsequently, the as-synthesized Co-2DP was deposited onto a Ti electrode and explored as electrocatalysts for HER in 1.0 M aqueous KOH solution (Fig. 5c) for a proof-of-concept application of the synthetic 2DP. The catalytic activity of the bare Ti electrode and cobalt-free 2DP (4) showed negligible HER activity. In contrast, the Co-2DP deposited on Ti substrate exhibited an excellent electrocatalytic activity for HER, suggesting the crucial active sites of Co-$N_4$ for HER. Specifically, the Co-2DP required an overpotential of about 367 mV to reach current density of 1.0 mA cm$^{-2}$; and an onset potential of 308 mV and a Tafel slope of 126 mV decade$^{-1}$ were derived from the polarization curve (Fig. 5d). These values are superior to those of the previously reported cobalt complex based molecular HER catalysts (Supplementary Table 1), including diimine-dioxime cobalt catalysts, cobalt microperoxidase-11and porphyrin-cobalt embedded in graphene oxide sheets[35,38,39].

## Discussion

In summary, we have demonstrated the synthesis of wafer-sized, 2D conjugated polymer with a monolayer thickness of ∼0.7 nm through Schiff-base polycondensation reaction at an air–water interface. The achieved monolayer 2DP has a Young's modulus of 267 ± 30 GPa, which is on the same order of graphene (200–1000 GPa). The 2DP has an optical band gap of 1.4 eV and can function as an active semiconducting layer in the TFT. Multilayer 2DP from cobalt-porphyrins catalyses hydrogen generation from water with a performance superior to those of cobalt-nitrogen (Co-$N_4$) coordination based molecular catalysts that grafted onto carbon nanotubes or immobilized in graphene oxide sheets. This work opens the door for interfacial synthesis of functional 2DPs using reversible polycondensation reaction, which may pave the way for the rational synthesis of 2D organic soft materials as promising candidates for next generation electronics and energy-related applications.

## Methods

**Materials.** 5, 10, 15, 20-Tetrakis (4-aminophenyl) porphyrin (monomer **1**) and 5, 10, 15, 20-Tetrakis (4-aminophenyl) porphyrin-Co(II) (monomer **2**) were obtained from TCI Deutschland, GmbH (Germany) and Porphyrin Laboratories GmbH (Germany), respectively. They were used as received. In all, 2, 5-dihydroxyterephthalaldehyde (monomer **3**) was synthesized according to previously reported procedure[40].

**Synthesis of monolayer 2DP.** The monolayer 2DP was prepared at an air–water interface of a Langmuir-Blodgett trough (Minitrough, KSV NIMA, Finland)[33]. The trough was equipped with a platinum Wilhelmy plate, a Taflon dipper and a pair of Delrin barriers. The substrates were immersed in to the subphase (water) and 30 µl of chloroform solution of monomer **1** (1 mg ml$^{-1}$) were spread on the water surface. The solvent was allowed to evaporate for 30 min and the compression was then done at a rate of 1 mm min$^{-1}$. When the surface pressure reached 10 mN m$^{-1}$, 2 ml of aqueous solution of monomer **3** (4.4 µmol l$^{-1}$) was added to the subphase. The molar amount of monomer **3** was ∼200-fold excess to that of monomer **1**. After 16 h of polymerization, the single layer 2DP was horizontally transferred onto substrates with a rate of 1 mm min$^{-1}$. The samples were rinsed with chloroform and water and then dried in N$_2$ flow.

**Synthesis of multilayer 2DP.** The multilayer was synthesized on a custom made glass trough[41]. Fifteen millilitres of chloroform solution of monomer **1** or **2** (1, 2 × 10$^{-4}$ mol l$^{-1}$) was provided in the glass trough, followed by overlaying 15 ml of an aqueous solution of monomer **3** (1 × 10$^{-3}$ mol l$^{-1}$). After the resulting interface was allowed to stay undisturbed for 10 days, the as-synthesized multilayer 2DP was fished onto substrates of interests for characterization.

**Fabrication of graphene-2DP-graphene sandwich structure.** CVD graphene flakes were transferred to a Quantifoil grid, followed by deposition of monolayer 2DP. In the next step, 2-3 µl of Milli-Q water was drop casted on the surface of the 2DP. Another Quantifoil grid with CVD graphene flakes was laid subsequently on top of the water droplet with the graphene flakes facing the 2DP. The grids were left in ambience for 24 h until the water was completely dried. As a result, a

graphene-2DP-graphene has been constructed. Since the electron diffraction experiments were conducted under cryogenic conditions, a thin layer of gold (approx. 20 nm) was sputtered onto the other side of the Quantifoil grids to improve thermal conductance.

**Preparation of PDMS.** PDMS precursor (Dow Corning Sylgard 184) was mixed in a 10:1 mass ratio of oligomeric base to curing agent, degassed for 20 min and poured into 100 × 100 mm poly(styrene) molds. The precursor was allowed to harden at room temperature for 48 h on a tared table followed by a thermal curing at 80 °C for 4 h. Eventually the hardened PDMS was cut in 55 × 10 × 5 mm slabs, washed carefully with Milli-Q-water and dried with N$_2$. The resulted PDMS has a Young's modulus and Poisson ratio of 1.85 MPa and 0.5, respectively[42].

**Preparation of monolayer 2DP on PDMS.** The monolayer was lifted out of the air–water interface of a LB trough using slabs of PDMS that has been treated before with 10% HCl. The samples were dried at 50 °C for 30 min and stretched by a pre-determined strain of 20%. Measuring the necking at this pre-strain revealed a substrate compression of 8.48%.

**2D Fourier transformation (2D-FT) analysis.** The wrinkle periodicity, that is, the wavelength, was determined using 2D-FT. The 2D-FT transformed image was integrated perpendicular to the periodic structures ($k_x$). The central zero-order peak between 0 and 2.5 µm$^{-1}$ was omitted. The information of the wrinkle periodicity was evaluated from the $k_x$ range between 2.5 and 15 µm$^{-1}$. The integration in $k_x$ resulted a peak maximum at 6.57 µm$^{-1}$, corresponding to a wavelength of 0.152 µm. Integration in $k_y$, that is, parallel to the wrinkles, yielded a shoulder at around 0.3 µm$^{-1}$, which most likely can be attributed to the average crack density of one crack per ∼3.3 µm of wrinkled area.

**Thin film transistor (TFT).** TFT structures on 230 nm SiO$_2$/Si with 30 nm thick Au patterns for source/drain contact were purchased from Fraunhofer IPMS, Germany. The length and width of the transistor channels are 20 and 500 µm, respectively. The single layer 2DP was horizontally transferred onto the substrate by Langmuir Schäfer method. Transfer curves of the TFT were characterized by a HP 4145B parameter analyser in a nitrogen glove box.

**Hydrogen evolution reaction (HER).** All electrochemical measurements were carried out with an electrochemical workstation (CHI 760E, CH Instruments, USA) in a conventional three-electrode cell system using a Pt wire as the counter electrode and an Ag/AgCl electrode as the reference electrode. The Ti foil with thickness of 0.25 mm was used as substrate and the working electrode. The multilayer cobalt-2DP were horizontally transferred on to the surface of the Ti substrate (1 × 1 cm). All potentials in this study were relative to that of a reversible hydrogen electrode. The potential difference between Ag/AgCl and reversible hydrogen electrode was determined based on a calibration measurement in a H$_2$-saturated electrolyte. The polarization curves were obtained in an Ar-saturated 1.0 M KOH electrolytes with a scan rate of 5 mV s$^{-1}$ at room temperature. The long-term durability was tested using chronopotentiometric and chronoamperometric measurements.

**Data availability.** The data that support the findings of this study are available from the corresponding authors on request.

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

## Acknowledgements

This work was financially supported by the ERC Grant 2DMATER, ESF Young Researcher Group 'GRAPHD' and the EC under the Graphene Flagship (no. CNECT-ICT-604391). B.A.G., C.K. and A.F. acknowledge financial support by the European Research Council under the European Union's Seventh Framework Program (FP/2007-2013)/ERC Grant Agreement No. METAMECH-306686. Z.Z. gratefully appreciates funding from Marie-Curie Fellowship (CONJUGATION-706082). We thank R. Jordan (TUD) for offering access to AFM and Raman spectroscopy. We acknowledge the use of the facilities in Dresden Center for Nanoanalysis (DCN) at TUD.

## Author contributions

Z.Z. and X.F. conceived the topic and experiments. H.S. carried out most of the experiments. H.Q., T.L. and U.K. performed TEM and SAED. B.A.G., C.K. and A.F. measured the mechanical properties of the monolayer 2DP. D.T. and G.S. conducted DFT calculation. R.D. for helpful discussion. Y.H. performed HER. T.Z. conducted AFM and Raman spectroscopy. Z.Z. and X.F. wrote the manuscript. All authors discussed the results and commented on the manuscript.

## Additional information

**Competing financial interests:** The authors declare no competing financial interests.

