## [Peer Review File · Nature Communications]

Reviewers' comments:

Reviewer #1 (Remarks to the Author):

Zheng and Feng et al. reported two polyimine-based 2D polymers by interfacial polymerization. This article described a substantial amount of interesting work. It is commendable that the authors tried to identify possible applications for the 2D polymers. However, the characterization of the 2D polymers is a little bit reckless. The authors could also have deepened their study and discussion in one promising area instead of superficially measuring several completely different properties including mechanical strength, conductivity, and catalysis in one communication. Thus, I recommend the acceptance of this manuscript after major revision.

Here are the comments and questions for revision:

- 1) Defects and/or possible crossing-linking play important roles in determining properties of the 2D polymers. Although the optical microscopy image of the monolayer 2D polymer showed a homogenous film, the scanning electron microscopy image with the same size showed a few cracks and folds (Figure 2a and b). Which one is a better representation of the 2D polymers, especially for the ones used in the property tests?
- 2) It was a little hasty to draw a conclusion about homogeneity of the 2D polymers using an optical microscopy image and a linear correlation between the absorbance value of the S band and the number of layers. More experimental evidences are needed to confirm the homogeneity, especially at molecular level, and the size of the 2D polymers
- 3) It is difficult to understand why the AFM image of 2D polymer had to be measured when the film was still wet. Why did water form a nice 0.3 nm layer to cover 2D polymer which contained mostly conjugated sp² hybridized C and N atoms (Figure 2c and 2e)? There are nano-sized pores within the 2D polymers (Figure 1 and 2e). Was it observed under AMF?
- 4) Some data in the Table 1S of the supporting information are consistent with the ones in a similar Table S1 of reference 34 published by Feng in 2015, but some are different. The authors need to double check the data and clarify it.

Reviewer #2 (Remarks to the Author):

This work reports the synthesis of two-dimensional conjugated polymers through imine condensation reaction at an air-water or liquid-liquid interface. The resulting polymers were characterized with a series of techniques, such as SEM, AFM, TEM, Raman, UV-Vis, XPS, etc. Theoretical calculation was also performed to model the polymer conformation and electronic band gap. The resulting monolayer 2D polymers exhibit high mechanical stiffness (Young's modulus of 100GPa) and multilayer 2DPs display electrocatalytic activity for water

reduction (hydrogen evolution). Given the growing interest in the study of 2DPs, particularly those monolayer ones, and the knowledge gained in this study that could help future development of novel 2DPs with good mechanical and/or electronic properties, publication of this work on Nature Communication is recommended. There are a few minor issues the referee would like to point out:

1. The following seminal review article on 2DPs should be added to ref 1-3: Angew. Chem. Int. Ed. 2009, 48, 1030.
2. The following review article on Schiff-base condensation reaction, particularly for COF synthesis should be added to ref 13: CrystEngComm 2013, 15, 1484.
3. Page 3, line 61, "without intermolecular hydrogen bond" should be "intramolecular"?
4. Page 3, line 61-63, in the work of ref 15, the crystallinity (structural order) of the covalent monolayer was not reported/discussed, so whether that 2DP is crystalline or not is still an open question. Therefore, it is not accurate to make the statement of "they lack crystallinity".
5. The following article on using Tip-Enhanced Raman Spectroscopy (TERS) to characterize 2DP should be added to Ref 20-21: ACS Nano. 2015, 9, 4252.
6. It is not very clear whether the regular confocal Raman (NT-MDT) or Surface-Enhanced Raman (SERS) was used to characterize the monolayer polymer. The authors should make it clear in the manuscript and supporting information.

Reviewer #3 (Remarks to the Author):

The paper reports 2D structures of cobalt which has been applied into energy, namely, Hydrogen production. While I find the structures useful and interesting, the novelty is not as high as the authors claim.

There are issues on benchmarking the HER, namely we have yet another material for the HER. While the LSV is given in figure 5, benchmarking to other 2D materials such as MoS₂ etc is not given nor Pt. In my estimation the HER is not that exciting. What also is the rate limiting step and ToF (see many papers which characterise the HER in more detail - this is needed here).

The work has potential to incorporate more interesting materials and applications. I do not find the work of such a high novelty to publish here.

Reviewers' comments:

Reviewer #1 (Remarks to the Author):

Zheng and Feng et al. reported two polyimine-based 2D polymers by interfacial polymerization. This article described a substantial amount of interesting work. It is commendable that the authors tried to identify possible applications for the 2D polymers. However, the characterization of the 2D polymers is a little bit reckless. The authors could also have deepened their study and discussion in one promising area instead of superficially measuring several completely different properties including mechanical strength, conductivity, and catalysis in one communication. Thus, I recommend the acceptance of this manuscript after major revision.

Response:

We greatly appreciate the reviewer for the positive comments and suggestion to deepen the study in one specific area. Based on the reviewer's suggestions, additional experiments have been performed and point-by-point responses to all the specific comments raised have been provided below. In the meantime, we would like to keep all these measurements (mechanical strength, conductivity and catalysis) in the current manuscript to provide more insights to the emerging class of materials - 2D polymers, which are still at an initial stage of development. The high mechanical strength of the 2DP (comparable to graphene and graphene oxide) tells that it can be of significant difference to traditional (linear) polymers. The conductivity measurement indicates that the 2DP can serve as a semiconductor, the basis of electronics and optoelectronics. The proof-of-concept application of the achieved 2DPs in electrocatalysis may provide the opportunity to understand how molecular electrocatalysts behave when they are connected in a 2D network.

1) Defects and/or possible crossing-linking play important roles in determining properties of the 2D polymers. Although the optical microscopy image of the monolayer 2D polymer showed a homogenous film, the scanning electron microscopy image with the same size showed a few cracks and folds (Figure 2a and b). Which one is a better representation of the 2D polymers, especially for the ones used in the property tests?

Response:

We greatly appreciate the valuable comments from the reviewer. We are sorry for the confusion caused regarding the optical microscopy and scanning electron microscopy (SEM) images of 2D polymers. Actually, a few cracks and folds in the SEM images are caused by manual transfer of the 2D polymers directly from air-water interface to a holey substrate, namely copper grids here, where both the mechanical force impacted by the hand and the capillary force of water can induce the fracture and folding of a film freely suspended over holes. To reduce rupture and folding of 2D materials (graphene, MoS₂, etc.) caused by transfer and drying, typically an organic protecting (or supporting) layer such as poly(methyl methacrylate)

is needed. Even with the supporting layer, it is challenging to obtain intact 2D materials without cracks and folding after the transfer process. Here, the 2D polymer has been transferred onto copper grids manually from air-water interface without any protecting layer. Thus, the 2D polymer freely suspended over copper grids without cracks and folds over 1 mm² is quite impressive. When the 2DP has been transferred onto a solid support, both the mechanical force and the capillary force of water are of less problem, and the resultant films can better represent their intrinsic features. For the optical microscopy images and the physical property tests of the 2D polymers, the samples are deposited on a solid substrate.

2) It was a little hasty to draw a conclusion about homogeneity of the 2D polymers using an optical microscopy image and a linear correlation between the absorbance value of the S band and the number of layers. More experimental evidences are needed to confirm the homogeneity, especially at molecular level, and the size of the 2D polymers.

Response:

We thank the reviewer for the constructive comments. We also agree with the reviewer that the determination of the homogeneity of the synthetic 2D polymer at the molecular level will be much more conclusive than by optical microscopy and UV-vis spectroscopy. The latter two techniques can only provide qualitative information about sample homogeneities. However, it remains high challenging so far to determine the molecular-resolution structure of synthetic 2D polymers, such as by means of scanning tunnelling microscopy (STM). Till now, there is only one article (*J. Am. Chem. Soc.* **137**, 3450-3453 (2015)), which reported the molecular structure of a 2D polymer synthesized at an air-water interface by STM. To be frank, we have tried hard in structural determination of the obtained 2D polymer by cooperation with STM experts. Unfortunately, no big success has been made so far. One of the main reasons is that the synthetic 2D polymer is not planar, which impedes a good contact with underlying conductive substrate and thus precludes molecular-resolution investigation with STM. Similar problem happened when we tried to study the 2DP using ultra-high vacuum (UHV) STM. We have performed thermal annealing at 120°C under both ambient and UHV conditions as well as solvent rinsing with isopropanol at room temperature to enhance the contact between the 2D polymer and STM substrates (such as highly oriented pyrolytic graphite (HOPG) and Au(111)) for the imaging, and the problem remain unsolved yet. Therefore, in the revised manuscript, we have referred the claim of homogeneity to macroscopic level.

Nevertheless, as the alternative, transmission electron microscopy (TEM) with the selected area electron diffraction (SAED) offers a means to probe the structural ordering of 2D polymers (and other 2D materials) at the unit-cell level. Since 2DPs are not stable under electron irradiation, we developed a new strategy to enhance the electron radiation resistance of the monolayer 2DP by sandwiching it between graphene layers (graphene encapsulation method), which enables structural characterization of individual organic layers by electron diffraction based technique. To further prove the homogeneity of the 2D polymer, we have analyzed the

graphene sandwiched structure by SAED over four samples. The experimental results are highly reproducible with all the samples prepared, similar SAED patterns can be obtained as far as the sandwich structure is employed, which indicates the homogeneity of the 2DP at the unit-cell level. In the Supplementary Information, we have added another 15 SAED images from the imaged samples in the revised manuscript. We believe that this developed SAED method can be also useful for the structure characterization of organic 2D materials with electron irradiation based techniques.

3) It is difficult to understand why the AFM image of 2D polymer had to be measured when the film was still wet. Why did water form a nice 0.3 nm layer to cover 2D polymer which contained mostly conjugated sp² hybridized C and N atoms (Figure 2c and 2e)? There are nano-sized pores within the 2D polymers (Figure 1 and 2e). Was it observed under AMF?

The 0.3 nm difference between the measured and actual thickness of the 2DP mostly originates from the surface roughness of the substrate (300 nm SiO₂/Si) and the tip-surface interaction during AFM measurement. Actually, similar phenomena have been observed in detecting the thickness of single layer graphene and other 2D materials. For instance, graphene has a real thickness of 0.34 nm, while the detected value on 300 nm SiO₂/Si appears to be 0.4 to 1.7 nm (Appl. Phys. Lett. 2014, 104, 171603; Nanotechnology, 2016, 27, 125704; Science, 2014, 306, 666-669). In the revised manuscript, we have deleted the sentence stating 0.3 nm of water layer, and the new sentence has been added “The crack helps to locate an AFM tip to measure the film thickness, which amounts to ~ 0.7 nm (Fig. 2c), thus suggesting the single-layer feature of the synthesized 2DP”.

Identifying the nano-sized pores has been one of the major challenges in the characterizations of synthesized 2D polymer. Unfortunately, no resolved pore structures by AFM has been reported or achieved for such systems thus far.

4) Some data in the Table 1S of the supporting information are consistent with the ones in a similar Table S1 of reference 34 published by Feng in 2015, but some are different. The authors need to double check the data and clarify it.

We are sorry for the caused confusion. With respect to reference 34, HER reaction was performed in acidic solution (0.5 M H₂SO₄), whereas in our work the HER reaction was performed in basic solution (1 M KOH). All the data have been doubled checked and mistakes have been corrected in the revised Table S1.

Reviewer #2 (Remarks to the Author):

This work reports the synthesis of two-dimensional conjugated polymers through imine condensation reaction at an air-water or liquid-liquid interface. The resulting polymers were

characterized with a series of techniques, such as SEM, AFM, TEM, Raman, UV-Vis, XPS, etc. Theoretical calculation was also performed to model the polymer conformation and electronic band gap. The resulting monolayer 2D polymers exhibit high mechanical stiffness (Young's modulus of 100GPa) and multilayer 2DPs display electrocatalytic activity for water reduction (hydrogen evolution). Given the growing interest in the study of 2DPs, particularly those monolayer ones, and the knowledge gained in this study that could help future development of novel 2DPs with good mechanical and/or electronic properties, publication of this work on Nature Communication is recommended. There are a few minor issues the referee would like to point out:

Response:

We greatly appreciate the valuable comments from the reviewer. All the suggestions from the reviewer have been carefully addressed and changes have been made accordingly.

1. The following seminal review article on 2DPs should be added to ref 1-3: Angew. Chem. Int. Ed. 2009, 48, 1030.

We thank the reviewer for pointing this out. Following the suggestion, we have added Angew. Chem. Int. Ed. 2009, 48, 1030 to reference 3 in the revised manuscript.

2. The following review article on Schiff-base condensation reaction, particularly for COF synthesis should be added to ref 13: CrystEngComm 2013, 15, 1484.

Following the suggestion from the reviewer, we have added CrystEngComm 2013, 15, 1484 to reference 14 in the revised manuscript.

3. Page 3, line 61, "without intermolecular hydrogen bond" should be "intramolecular"?

Typing error 'without intermolecular hydrogen bond' has been changed to "without intramolecular hydrogen bond" in the revised manuscript.

4. Page 3, line 61-63, in the work of ref 15, the crystallinity (structural order) of the covalent monolayer was not reported/discussed, so whether that 2DP is crystalline or not is still an open question. Therefore, it is not accurate to make the statement of "they lack crystallinity".

We thank the reviewer for pointing out this. We have replaced the statement "they lack crystallinity" with "they have no proven crystallinity" in the revised manuscript.

5. The following article on using Tip-Enhanced Raman Spectroscopy (TERS) to characterize 2DP should be added to Ref 20-21: ACS Nano. 2015, 9, 4252.

The referred literature (ACS Nano. 2015, 9, 4252) has been added in the revised manuscript.

6. It is not very clear whether the regular confocal Raman (NT-MDT) or Surface-Enhanced Raman (SERS) was used to characterize the monolayer polymer. The authors should make it clear in the manuscript and supporting information.

We used confocal Raman spectroscopy, which has been included in the revised manuscript

(line 158).

Reviewer #3 (Remarks to the Author):

The paper reports 2D structures of cobalt which has been applied into energy, namely, Hydrogen production. While I find the structures useful and interesting, the novel is not as high as the authors claim.

There are issues on benchmarking the HER, namely we have yet another material for the HER. While the LSV is given in figure 5, benchmarking to other 2D materials such as MoS₂ etc is not given nor Pt. In my estimation the HER is not that exciting. What also is the rate limiting step and ToF (see many papers which characterise the HER in more detail - this is needed here).

The work has potential to incorporate more interesting materials and applications. I do not find the work of such a high novelty to publish here.

Response:

We appreciate the valuable comments from the reviewer. Probably the reviewer has misunderstood the major achievement in the current work. Actually our work has been focused on developing a novel interfacial synthesis of two-dimensional conjugated polymers through imine condensation reaction with high mechanical stiffness and implementing new characterization techniques. The study of HER for the obtained cobalt-based 2D polymers is just to show one of their proof-of-concept functions. On the other hand, the achieved 2D polymer (2DP) for HER may provide the possibility to gain fundamental understanding of how molecular catalysts behave when they are connected in a 2D network. We did not aim to achieve the state-of-the-art HER performance for the developed 2D polymers.

Reviewers' comments:

Reviewer #1 (Remarks to the Author):

Zheng and Feng et al. have addressed part of the concerns of the reviewers. The manuscript has been improved after revision. However, the characterization of the 2D polymers is still rather weak, especially considering the strong conclusions of the paper. Thus, I recommend the acceptance of this manuscript after revision.

Here are some additional comments and suggestions for revision:

The authors reported an excellent Young's modulus number of their 2D polymer (297 GPa), which was much higher than that of graphene oxide (208 GPa, Ref. 32). The result is a little bit unexpected if one compares the chemical structures of the 2D polymer in this manuscript and graphene oxide. In the Results, Conclusion and Supporting Information, the authors claimed that their monolayer 2D polymer had a Young's modulus of 297 GPa. In the abstract, the authors said the Young's modulus was in excess of 100 GPa. Although the two statements were not contradictory, the numbers were quite different. To minimize measurement error, the Young's modulus should be measured multiple times and the average together with all the raw data should be included in the paper. It is also important to run a control by measuring a material such as graphene oxide or graphene with known Young's modulus using the same method under the same conditions. The results of the control experiment should also be included in the manuscript or supporting information.

Reviewer #3 (Remarks to the Author):

OK, I understand the HER work was not the main focus, then in the abstract you should remove the Tafel data and in the main text, you should point this out that this work was just proof of concept.

Reviewers' comments:

Reviewer #1 (Remarks to the Author):

Zheng and Feng et al. have addressed part of the concerns of the reviewers. The manuscript has been improved after revision. However, the characterization of the 2D polymers is still rather weak, especially considering the strong conclusions of the paper. Thus, I recommend the acceptance of this manuscript after revision.

Response:

We greatly appreciate the reviewer for the positive comments. We agree with the reviewer that current selected area electron diffraction pattern cannot tell the molecular structure of the 2DPs beyond unit-cell level. Therefore, we have weakened the claim of the structure in the revised manuscript by changing the sentence in line 120 – 122 from “It is worth mentioning that the same diffraction pattern has been repeatedly observed from different positions of the monolayer, which indicates high crystallinity of the synthesized 2DP.” to “It is worth mentioning that the same diffraction rings have been repeatedly observed from different positions of the monolayer, which indicates good homogeneity of the polycrystalline 2DP at the unit-cell level.”. As atomic and molecular level imaging of synthetic 2DPs is still a major challenge, we hope that our work will further inspire efforts in this emerging research field. Based on the reviewer’s suggestions, additional experiments have been performed and point-by-point responses to all the specific comments raised have been provided below.

Here are some additional comments and suggestions for revision: The authors reported an excellent Young's modulus number of their 2D polymer (297 GPa), which was much higher than that of graphene oxide (208 GPa, Ref. 32). The result is a little bit unexpected if one compares the chemical structures of the 2D polymer in this manuscript and graphene oxide. In the Results, Conclusion and Supporting Information, the authors claimed that their monolayer 2D polymer had a Young's modulus of 297 GPa. In the abstract, the authors said the Young's modulus was in excess of 100 GPa. Although the two statements were not contradictory, the numbers were quite different. To minimize measurement error, the Young's modulus should be measured multiple times and the average together with all the raw data should be included in the paper. It is also important to run a control by measuring a material such as graphene oxide or graphene with known Young's modulus using the same method under the same conditions. The results of the control experiment should also be included in the manuscript or supporting information.

Response:

We greatly appreciate the constructive comments from the reviewer. We have measured multiple times of the Young's modulus of the 2D polymers based on different samples; the results are highly reproducible, and the average value with an error bar has been included in the revised manuscript. All the raw data have been added in the Supplementary Information. In this work, we used wrinkling method for the evaluation of the Young's modulus of 2D polymer. This method has been widely explored to determine the mechanical stiffness of various materials, such as thin polymeric films (Nat. Mater. 2004, 3, 545–550; Ref. 28), clay

(ACS Appl. Mater. Interfaces 2013, 5, 5851–5855), graphene and graphene oxide (Adv. Mater. 2013, 25, 1337–1341). Note that the measurement of the Young's modulus of the graphene and graphene oxide in *Adv. Mater.* 2013, 25, 1337–1341 has been done with exactly the same method under same conditions as in the current work. The measured values for graphene (0.91 ± 0.07 Tpa) and graphene oxide (0.23 ± 0.07 Tpa) are comparable to those determined by AFM nanoindentation (1Tpa for graphene, Ref. 31) and contact-force-measurements (0.21 ± 0.02 Tpa for graphene oxide, Ref 32). To further confirm the generality and reliability of this method, we also measured the Young's modulus of a bis(terpyridine)-metal complex nanosheet reported by Schlüter et al with this method, which gave similar result to that determined by AFM nano-indentation (Ref. 33).

Reviewer #3 (Remarks to the Author):

OK, i understand the HER work was not the main focus, then in the abstract you should remove the tafel data and in the main text, you should point this out that this work was just proof of concept.

Response:

We greatly appreciate the valuable comments from the reviewer. Accordingly, we have deleted the Tafel data in the abstract and pointed out that this work was just a proof-of-concept application of the synthetic 2D polymers in the main text (page 12, marked in red).

REVIEWERS' COMMENTS:

Reviewer #1 (Remarks to the Author):

I will recommend the acceptance of this manuscript.

Point-to-point response to the reviewers' comments

Reviewers' comments:

Reviewer #1 (Remarks to the Author):

I will recommend the acceptance of this manuscript.

Response:

We greatly appreciate the reviewer for the positive comment.